# Inference in Deep Gaussian Processes using Stochastic Gradient Hamiltonian Monte Carlo

**Marton Havasi**
Department of Engineering
University of Cambridge
mh740@cam.ac.uk

**José Miguel Hernández-Lobato**
Department of Engineering
University of Cambridge,
Microsoft Research,
Alan Turing Institute
jmh233@cam.ac.uk

**Juan José Murillo-Fuentes**
Department of Signal Theory and Communications
University of Sevilla
murillo@us.es

## Abstract

Deep Gaussian Processes (DGPs) are hierarchical generalizations of Gaussian Processes that combine well calibrated uncertainty estimates with the high flexibility of multilayer models. One of the biggest challenges with these models is that exact inference is intractable. The current state-of-the-art inference method, Variational Inference (VI), employs a Gaussian approximation to the posterior distribution. This can be a potentially poor unimodal approximation of the generally multimodal posterior. In this work, we provide evidence for the non-Gaussian nature of the posterior and we apply the Stochastic Gradient Hamiltonian Monte Carlo method to generate samples. To efficiently optimize the hyperparameters, we introduce the Moving Window MCEM algorithm. This results in significantly better predictions at a lower computational cost than its VI counterpart. Thus our method establishes a new state-of-the-art for inference in DGPs.

## 1   Introduction

Deep Gaussian Processes (DGP) [Damianou and Lawrence, 2013] are multilayer predictive models that are highly flexible and can accurately model uncertainty. In particular, they have been shown to perform well on a multitude of supervised regression tasks ranging from small ($\sim$500 datapoints) to large datasets ($\sim$500,000 datapoints) [Salimbeni and Deisenroth, 2017, Bui et al., 2016, Cutajar et al., 2016]. Their main benefit over neural networks is that they are capable of capturing uncertainty in their predictions. This makes them good candidates for tasks where the prediction uncertainty plays a crucial role, for example, black-box Bayesian Optimization problems and a variety of safety-critical applications such as autonomous vehicles and medical diagnostics.

Deep Gaussian Processes introduce a multilayer hierarchy to Gaussian Processes (GP) [Williams and Rasmussen, 1996]. A GP is a non-parametric model that assumes a jointly Gaussian distribution for any finite set of inputs. The covariance of any pair of inputs is determined by the covariance function. GPs can be a robust choice due to being non-parametric and analytically computable, however, one issue is that choosing the covariance function often requires hand tuning and expert knowledge of the dataset, which is not possible without prior knowledge of the problem at hand. In a multilayer hierarchy, the hidden layers overcome this limitation by stretching and warping the input space,

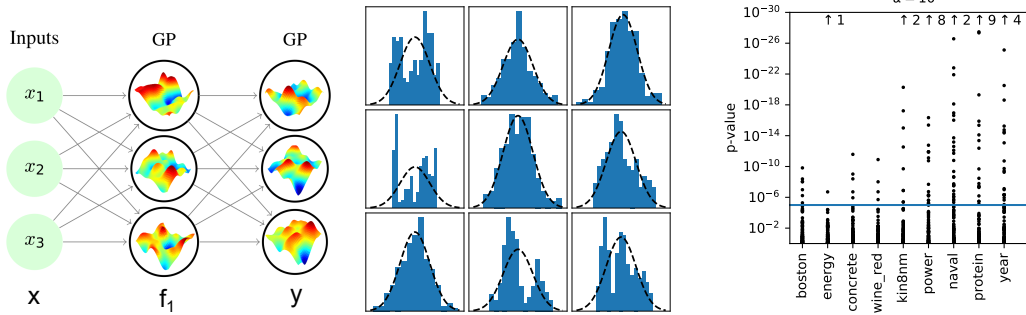

Figure 1: (Left): Deep Gaussian Process illustration[1]. (Middle): Histograms of a random selection of inducing outputs. The best-fit Gaussian distribution is denoted with a dashed line. Some of them exhibit a clear multimodal behaviour. (Right): P-values for 100 randomly selected inducing outputs per dataset. The null hypotheses are that their distributions are Gaussian.

resulting in a Bayesian 'self-tuning' covariance function that fits the data without any human input [Damianou, 2015].

The deep hierarchical generalization of GPs is done in a fully connected, feed-forward manner. The outputs of the previous layer serve as an input to the next. However, a significant difference from neural networks is that the layer outputs are probabilistic rather than exact values so the uncertainty is propagated through the network. The left part of Figure 1 illustrates the concept with a single hidden layer. The input to the hidden layer is the input data $x$ and the output of the hidden layer $f_1$ serves as the input data to the output layer, which itself is formed by GPs.

Exact inference is infeasible in GPs for large datasets due to the high computational cost of working with the inverse covariance matrix. Instead, the posterior is approximated using a small set of pseudo datapoints ($\sim$100) also referred to as inducing points [Snelson and Ghahramani, 2006, Titsias, 2009, Quiñonero-Candela and Rasmussen, 2005]. We assume this inducing point framework throughout the paper. Predictions are made using the inducing points to avoid computing the covariance matrix of the whole dataset. Both in GPs and DGPs, the inducing outputs are treated as latent variables that need to be marginalized.

The current state-of-the-art inference method in DGPs is Doubly Stochastic Variation Inference (DSVI) [Salimbeni and Deisenroth, 2017] which has been shown to outperform Expectation Propagation [Minka, 2001, Bui et al., 2016] and it also has better performance than Bayesian Neural Networks with Probabilistic Backpropagation [Hernández-Lobato and Adams, 2015] and Bayesian Neural Networks with earlier inference methods such as Variation Inference [Graves, 2011], Stochastic Gradient Langevin Dynamics [Welling and Teh, 2011] and Hybrid Monte Carlo [Neal, 1993]. However, a drawback of DSVI is that it approximates the posterior distribution with a Gaussian. We show, with high confidence, that the posterior distribution is non-Gaussian for every dataset that we examine in this work. This finding motivates the use of inference methods with a more flexible posterior approximations.

In this work, we apply an inference method new to DGPs, Stochastic Gradient Hamiltonian Monte Carlo (SGHMC), a sampling method that accurately and efficiently captures the posterior distribution. In order to apply a sampling-based inference method to DGPs, we have to tackle the problem of optimizing the large number of hyperparameters. To address this problem, we propose Moving Window Monte Carlo Expectation Maximization, a novel method for obtaining the Maximum Likelihood (ML) estimate of the hyperparameters. This method is fast, efficient and generally applicable to any probabilistic model and MCMC sampler.

One might expect a sampling method such as SGHMC to be more computationally intensive than a variational method such as DSVI. However, in DGPs, sampling from the posterior is inexpensive, since it does not require the recomputation of the inverse covariance matrix, which only depends on

the hyperparameters. Furthermore, calculating the layerwise variance has a higher cost in the VI setting.

Lastly, we conduct experiments on a variety of supervised regression and classification tasks. We show empirically that our work significantly improves predictions on medium-large datasets at a lower computational cost.

Our contributions can be summarized in three points.

1. Demonstrating the non-Gaussianity of the posterior. We provide evidence that every regression dataset that we examine in this work has a non-Gaussian posterior.
2. We use SGHMC to directly sample from the posterior distribution of a DGP. Experiments show that this new inference method outperforms preceding works.
3. We introduce Moving Window MCEM, a novel algorithm for efficiently optimizing the hyperparameters when using a MCMC sampler for inference.

## 2   Background and Related Work

This section provides the background on Gaussian Processes and Deep Gaussian Processes for regression and establishes the notation used in the paper.

### 2.1   Single Layer Gaussian Process

Gaussian processes define a posterior distribution over functions $f : \mathbb{R}^D \to \mathbb{R}$ given a set of input-output pairs $\boldsymbol{x} = \{x_1, \ldots, x_N\}$ and $\boldsymbol{y} = \{y_1, \ldots, y_N\}$ respectively. Under the GP model, it is assumed that the function values $\boldsymbol{f} = f(\boldsymbol{x})$, where $f(\boldsymbol{x})$ denotes $\{f(x_1), \ldots, f(x_N)\}$, are jointly Gaussian with a fixed covariance function $k : \mathbb{R}^D \times \mathbb{R}^D \to \mathbb{R}$. The conditional distribution of $\boldsymbol{y}$ is obtained via the likelihood function $p(\boldsymbol{y}|\boldsymbol{f})$. A commonly used likelihood function is $p(\boldsymbol{y}|\boldsymbol{f}) = \mathcal{N}(\boldsymbol{y}|\boldsymbol{f}, \boldsymbol{I}\sigma^2)$ (constant Gaussian noise).

The computational cost of exact inference is $O(N^3)$, rendering it computationally infeasible for large datasets. A common approach uses a set of pseudo datapoints $\boldsymbol{Z} = \{z_1, \ldots z_M\}$, $\boldsymbol{u} = f(\boldsymbol{Z})$ [Snelson and Ghahramani, 2006, Titsias, 2009] and writes the joint probability density function as

$$p(\boldsymbol{y}, \boldsymbol{f}, \boldsymbol{u}) = p(\boldsymbol{y}|\boldsymbol{f})p(\boldsymbol{f}|\boldsymbol{u})p(\boldsymbol{u})\,.$$

The distribution of $\boldsymbol{f}$ given the inducing outputs $\boldsymbol{u}$ can be expressed as $p(\boldsymbol{f}|\boldsymbol{u}) = \mathcal{N}(\boldsymbol{\mu}, \boldsymbol{\Sigma})$ with

$$\boldsymbol{\mu} = K_{\boldsymbol{x}\boldsymbol{Z}}K_{\boldsymbol{Z}\boldsymbol{Z}}^{-1}\boldsymbol{u}$$
$$\boldsymbol{\Sigma} = K_{\boldsymbol{x}\boldsymbol{x}} - K_{\boldsymbol{x}\boldsymbol{Z}}K_{\boldsymbol{Z}\boldsymbol{Z}}^{-1}K_{\boldsymbol{x}\boldsymbol{Z}}^T$$

where the notation $K_{AB}$ refers to the covariance matrix between two sets of points $A$, $B$ with entries $[K_{AB}]_{ij} = k(A_i, B_j)$ where $A_i$ and $B_j$ are the $i$-th and $j$-th elements of $A$ and $B$ respectively.

In order to obtain the posterior of $\boldsymbol{f}$, $\boldsymbol{u}$ must be marginalized, yielding the equation

$$p(\boldsymbol{f}|\boldsymbol{y}) = \int p(\boldsymbol{f}|\boldsymbol{u})p(\boldsymbol{u}|\boldsymbol{y})d\boldsymbol{u}\,.$$

Note that $\boldsymbol{f}$ is conditionally independent of $\boldsymbol{y}$ given $\boldsymbol{u}$.

For single layer GPs, Variational Inference (VI) can be used for marginalization. VI approximates the joint posterior distribution $p(\boldsymbol{f}, \boldsymbol{u}|\boldsymbol{y})$ with the variational posterior $q(\boldsymbol{f}, \boldsymbol{u}) = p(\boldsymbol{f}|\boldsymbol{u})q(\boldsymbol{u})$, where $q(\boldsymbol{u}) = \mathcal{N}(\boldsymbol{u}|\boldsymbol{m}, \boldsymbol{S})$.

This choice of $q(\boldsymbol{u})$ allows for exact inference of the marginal $q(\boldsymbol{f}|\boldsymbol{m}, \boldsymbol{S}) = \int p(\boldsymbol{f}|\boldsymbol{u})q(\boldsymbol{u})d\boldsymbol{u} = \mathcal{N}(\boldsymbol{f}|\tilde{\mu}, \tilde{\Sigma})$

$$\begin{aligned} \text{where } \tilde{\mu} &= K_{\boldsymbol{x}\boldsymbol{Z}}K_{\boldsymbol{Z}\boldsymbol{Z}}^{-1}\boldsymbol{m}\,, \\ \tilde{\Sigma} &= K_{\boldsymbol{x}\boldsymbol{x}} - K_{\boldsymbol{x}\boldsymbol{Z}}K_{\boldsymbol{Z}\boldsymbol{Z}}^{-1}(K_{\boldsymbol{Z}\boldsymbol{Z}} - \boldsymbol{S})K_{\boldsymbol{Z}\boldsymbol{Z}}^{-1}K_{\boldsymbol{x}\boldsymbol{Z}}^T\,. \end{aligned} \tag{1}$$

The variational parameters $\boldsymbol{m}$ and $\boldsymbol{S}$ need to be optimized. This is done by minimizing the Kullback-Leibler divergence of the true and the approximate posteriors, which is equivalent to maximizing a lower bound to the marginal likelihood (Evidence Lower Bound or ELBO):

$$\log p(\boldsymbol{y}) \geq \mathbb{E}_{q(\boldsymbol{f}, \boldsymbol{u})}\big[\log p(\boldsymbol{y}, \boldsymbol{f}, \boldsymbol{u}) - \log q(\boldsymbol{f}, \boldsymbol{u})\big] = \mathbb{E}_{q(\boldsymbol{f}|\boldsymbol{m}, \boldsymbol{S})}\big[\log p(\boldsymbol{y}|\boldsymbol{f})\big] - \mathrm{KL}\big[q(\boldsymbol{u})||p(\boldsymbol{u})\big]\,.$$

## 2.2 Deep Gaussian Process

In a DGP of depth $L$, each layer is a GP that models a function $f_l$ with input $\boldsymbol{f}_{l-1}$ and output $\boldsymbol{f}_l$ for $l = 1, \ldots, L$ ($\boldsymbol{f}_0 = \boldsymbol{x}$) as illustrated in the left part of Figure 1. The inducing inputs for the layers are denoted by $\boldsymbol{Z}_1, \ldots, \boldsymbol{Z}_L$ with associated inducing outputs $\boldsymbol{u}_1 = f_1(\boldsymbol{Z}_1), \ldots, \boldsymbol{u}_L = f_L(\boldsymbol{Z}_L)$.

The joint probability density function can be written analogously to the GP model case:

$$p(\boldsymbol{y}, \{\boldsymbol{f}_l\}_{l=1}^L, \{\boldsymbol{u}_l\}_{l=1}^L) = p(\boldsymbol{y}|\boldsymbol{f}_L) \prod_{l=1}^L p(\boldsymbol{f}_l|\boldsymbol{u}_l) p(\boldsymbol{u}_l) \,. \tag{2}$$

## 2.3 Inference

The goal of inference is to marginalize the inducing outputs $\{\boldsymbol{u}_l\}_{l=1}^L$ and layer outputs $\{\boldsymbol{f}_l\}_{l=1}^L$ and approximate the marginal likelihood $p(\boldsymbol{y})$. This section discusses prior works regarding inference.

**Doubly Stochastic Variation Inference**   DSVI is an extension of Variational Inference to DGPs [Salimbeni and Deisenroth, 2017] that approximates the posterior of the inducing outputs $\boldsymbol{u}_l$ with independent multivariate Gaussians $q(\boldsymbol{u}_l) = \mathcal{N}(\boldsymbol{u}_l|\boldsymbol{m}_l, \boldsymbol{S}_l)$.

The layer outputs naturally follow the single layer model in Eq. 1:

$$q(\boldsymbol{f}_l|\boldsymbol{f}_{l-1}) = \mathcal{N}(\boldsymbol{f}_l|\tilde{\mu}_l, \tilde{\Sigma}_l) \,,$$

$$q(\boldsymbol{f}_L) = \int \prod_{l=1}^L q(\boldsymbol{f}_l|\boldsymbol{f}_{l-1}) d\boldsymbol{f}_l \ldots d\boldsymbol{f}_{L-1} \,.$$

The first term in the resulting ELBO, $\mathcal{L} = \mathbb{E}_{q(\boldsymbol{f}_L)}\big[\log p(\boldsymbol{y}|\boldsymbol{f}_L)\big] - \sum_{l=1}^L \mathrm{KL}\big[q(\boldsymbol{u}_l)||p(\boldsymbol{u}_l)\big]$, is then estimated by sampling the layer outputs through minibatches to allow scaling to large datasets.

**Sampling-based inference for Gaussian Processes**   In a related work, Hensman et al. [2015] use Hybrid MC sampling in single layer GPs. They consider joint sampling of the GP hyperparameters and the inducing outputs. This work cannot straightforwardly be extended to DGPs because of the high cost of sampling the GP hyperparameters. Moreover, it uses a costly method, Bayesian Optimization, to tune the parameters of the sampler which further limits its applicability to DGPs.

## 3   Analysis of the Deep Gaussian Process Posterior

Adopting a new inference method over variational inference is motivated by the restrictive form that VI assumes about the posterior distribution. The variational assumption is that $p(\{\boldsymbol{u}_l\}_{l=1}^L|\boldsymbol{y})$ takes the form of a multivariate Gaussian that assumes independence between the layers. While in a single layer model, a Gaussian approximation to the posterior is provably correct [Williams and Rasmussen, 1996], this is not the case for DGPs.

First, we illustrate with a toy problem that the posterior distribution in DGPs can be multimodal. Following that, we provide evidence that every regression dataset that we consider in this work results in a non-Gaussian posterior distribution.

**Multimodal toy problem**   The multimodality of the posterior of a two layer DGP ($L = 2$) is demonstrated on a toy problem (Table 1). For the purpose of the demonstration, we made the simplifying assumption that $\sigma^2 = 0$, so the likelihood function has no noise. This toy problem has two Maximum-A-Posteriori (MAP) solutions (Mode A and Mode B). The table shows the variational posterior at each layer for DSVI. We can see that it fits one of the modes randomly (depending on the initialization) while completely ignoring the other. On the other hand, a sampling method such as SGHMC (as implemented in the following section) explores both of the modes and therefore provides a better approximation to the posterior.

**Empirical evidence**   To further support our claim regarding the multimodality of the posterior, we give empirical evidence that ,for real-world datasets, the posterior is not Gaussian.

Table 1: The layer inputs and outputs of a two layer DGP. Under DSVI, we show the mean and the standard deviation of the variational distribution. In the case of SGHMC, samples from each layer are shown. The two MAP solutions are shown under Mode A and Mode B.

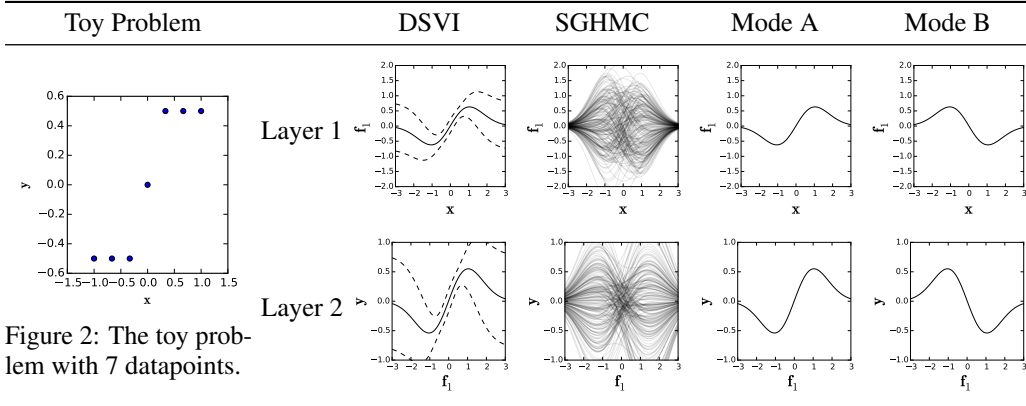

Figure 2: The toy problem with 7 datapoints.

We conduct the following analysis. Consider the null hypothesis that the posterior under a dataset is a multivariate Gaussian distribution. This null hypothesis implies that the distribution of each inducing output is a Gaussian. We examine the approximate posterior samples generated by SGHMC for each inducing output, using the implementation of SGHMC for DGPs described in the next section. In order to derive p-values, we apply the kurtosis test for Gaussianity [Cramer, 1998]. This test is commonly used to identify multimodal distributions because these often have significantly higher kurtosis (also called 4th moment).

For each dataset, we calculate the p-values of 100 randomly selected inducing outputs and compare the results against the probability threshold $\alpha = 10^{-5}$. The Bonferroni correction was applied to $\alpha$ to account for the high number of concurrent hypothesis tests. The results are displayed in the right part of Figure 1. Since every single dataset had p-values under the threshold, we can state with 99% certainty that all of these datasets have a non-Gaussian posterior.

## 4 Sampling-based Inference for Deep Gaussian Processes

Unlike with VI, when using sampling methods, we do not have access to an approximate posterior distribution $q(\boldsymbol{u})$ to generate predictions with. Instead, we have to rely on approximate samples generated from the posterior which in turn can be used to make predictions [Dunlop et al., 2017, Hoffman, 2017].

In practice, run a sampling process which has two phases. The burn-in phase is used to determine the hyperparameters of the model and the sampler. The hyperparameters of the sampler are selected using a heuristic auto-tuning approach, while the hyperparameters of the DGP are optimized using the novel Moving Window MCEM algorithm.

In the sampling phase, the sampler is run using the fixed hyperparameters. Since consecutive samples are highly correlated, we save one sample every 50 iterations and generate 200 samples for prediction.

Once the posterior samples are obtained, predictions can be made by combining the per-sample predictions to obtain a mixture distribution. Note that it is not more expensive to make predictions using this sampler than in DSVI since DSVI needs to sample the layer outputs to make predictions.

### 4.1 Stochastic Gradient Hamiltonian Monte Carlo

SGHMC [Chen et al., 2014] is a Markov Chain Monte Carlo sampling method [Neal, 1993] for producing samples from the intractable posterior distribution of the inducing outputs $p(\boldsymbol{u}|\boldsymbol{y})$ purely from stochastic gradient estimates.

With the introduction of an auxiliary variable, $\boldsymbol{r}$, the sampling procedure provides samples from the joint distribution $p(\boldsymbol{u}, \boldsymbol{r}|\boldsymbol{y})$. The equations that describe the MCMC process can be related to

**Algorithm 1:** Moving Window MCEM

```
initialize(θ);
initialize(u);
initialize(samples [1 ⋯ m]);
for i ← 0 to maxiter do
    u' ← randomElement(samples);
    stepSGD(∂p(y,u'|x,θ)/∂θ);
    u ~ p(u|y, x, θ));
    push_front(samples, u);
    pop_back(samples);
end
```

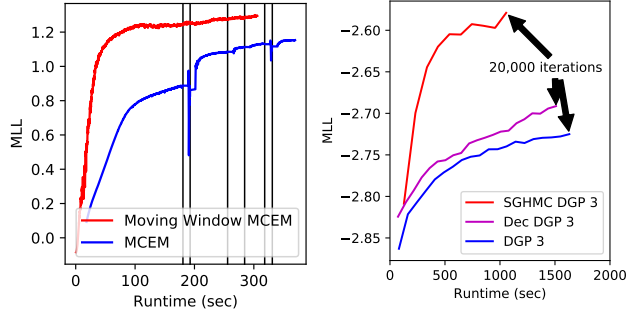

Figure 3: (Left): Pseudocode for Moving Window MCEM. (Middle): Comparison of predictive performance of Moving Window MCEM and MCEM algorithms. Vertical lines denote E-steps in MCEM algorithm. Higher is better. (Right): Comparison of the convergence of the different inference methods. Higher is better.

Hamiltonian dynamics [Brooks et al., 2011, Neal, 1993]. The negative log-posterior $U(\boldsymbol{u})$ acts as the potential energy and $\boldsymbol{r}$ plays the role of the kinetic energy:

$$p(\boldsymbol{u}, \boldsymbol{r}|\boldsymbol{y}) \propto \exp\big(-U(\boldsymbol{u}) - \frac{1}{2}\boldsymbol{r}^T M^{-1}\boldsymbol{r}\big),$$
$$U(\boldsymbol{u}) = -\log p(\boldsymbol{u}|\boldsymbol{y}).$$

In HMC the exact description of motion requires the computation of the gradient $\nabla U(\boldsymbol{u})$ in each update step, which is impractical for large datasets because of the high cost of integrating the layer outputs out in Eq. 2. This integral can be approximated by a lower bound that can be evaluated by Monte Carlo sampling [Salimbeni and Deisenroth, 2017]:

$$\log p(\boldsymbol{u}, \boldsymbol{y}) = \log \int p(\boldsymbol{y}, \boldsymbol{f}, \boldsymbol{u}) d\boldsymbol{f} \geq \int \log\left[\frac{p(\boldsymbol{y}, \boldsymbol{f}, \boldsymbol{u})}{p(\boldsymbol{f}|\boldsymbol{u})}\right] p(\boldsymbol{f}|\boldsymbol{u}) d\boldsymbol{f} \approx \log\left[\frac{p(\boldsymbol{y}, \boldsymbol{f}^i, \boldsymbol{u})}{p(\boldsymbol{f}^i|\boldsymbol{u})}\right],$$

where $\boldsymbol{f}^i$ is a Monte Carlo sample from the predictive distribution of the layer outputs: $\boldsymbol{f}^i \sim p(\boldsymbol{f}|\boldsymbol{u}) = \prod_{l=1}^{L} p(\boldsymbol{f}_l|\boldsymbol{u}_l, \boldsymbol{f}_{l-1})$. This leads to the estimate

$$\log p(\boldsymbol{u}, \boldsymbol{y}) \approx \log[p(\boldsymbol{y}|\boldsymbol{f}^i, \boldsymbol{u})p(\boldsymbol{u})] = \log p(\boldsymbol{y}|\boldsymbol{f}^i, \boldsymbol{u}) + \log p(\boldsymbol{u}),$$

that we can use to approximate the gradient since $\nabla U(\boldsymbol{u}) = -\nabla \log p(\boldsymbol{u}|\boldsymbol{y}) = -\nabla \log p(\boldsymbol{u}, \boldsymbol{y})$.

Chen et al. [2014] show that approximate posterior sampling is still possible with stochastic gradient estimates (obtained by subsampling the data) if the following update equations are used:

$$\Delta\boldsymbol{u} = \epsilon M^{-1}\boldsymbol{r},$$
$$\Delta\boldsymbol{r} = -\epsilon\nabla U(\boldsymbol{u}) - \epsilon C M^{-1}\boldsymbol{r} + \mathcal{N}\big(0, 2\epsilon(C - \hat{B})\big),$$

where $C$ is the friction term, $M$ is the mass matrix, $\hat{B}$ is the Fisher information matrix and $\epsilon$ is the step-size.

One caveat of SGHMC is that it has multiple parameters $(C, M, \hat{B}, \epsilon)$ that can be difficult to set without prior knowledge of the model and the data. We use the auto-tuning approach of Springenberg et al. [2016] to set these parameters which has been shown to work well for Bayesian Neural Networks (BNN). The similar nature of DGPs and BNNs strongly suggests that the same methodology is applicable to DGPs.

## 4.2 Moving Window Markov Chain Expectation Maximization

Optimizing the hyperparameters $\theta$ (parameters of the covariance function, inducing inputs and parameters of the likelihood function) prove difficult for MCMC methods [Turner and Sahani, 2011]. The naive approach consisting in optimizing them as the sampler progresses fails because

subsequent samples are highly correlated and as a result, the hyperparameters simply fit this moving, point-estimate of the posterior.

Monte Carlo Expectation Maximization (MCEM) [Wei and Tanner, 1990] is the natural extension of the Expectation Maximization algorithm that works with posterior samples to obtain the Maximum Likelihood estimate of the hyperparameters. MCEM alternates between two steps. The E-step samples from the posterior and the M-step maximizes the average log joint probability of the samples and the data:

E-step:                                M-step:

$$\boldsymbol{u}_{1\ldots m} \sim p(\boldsymbol{u}|\boldsymbol{y}, \boldsymbol{x}, \theta) \,. \qquad\qquad\qquad\qquad \theta = \arg\max_{\theta} Q(\theta) \,,$$

where $Q(\theta) = \frac{1}{m} \sum_{i=1}^{m} \log p(\boldsymbol{y}, \boldsymbol{u}_i | \boldsymbol{x}, \theta)$.

However, there is a significant drawback to MCEM: If the number of samples $m$ used in the M-step is too low then there is a risk of the hyperparameters overfitting to those samples. On the other hand, if $m$ is too high, the M-step becomes too expensive to compute. Furthermore, in the M-step, $\theta$ is maximized via gradient ascent, which means that the computational cost increases linearly with $m$.

To address this, we introduce a novel extension of MCEM called Moving Window MCEM. Our method optimizes the hyperparameters at the same cost as the previously described naive approach while avoiding its overfitting issues.

The idea behind Moving Window MCEM is to intertwine the E and M steps. Instead of generating new samples and then maximizing $Q(\theta)$ until convergence, we maintain a set of samples and take small steps towards the maximum of $Q(\theta)$. In the E-step, we generate one new sample and add it to the set while discarding the oldest sample (hence Moving Window). This is followed by the M-step, in which we take a random sample from the set and use it to take an approximate gradient step towards the maximum of $Q(\theta)$. Algorithm 1 on the left side of Figure 3 presents the pseudocode for Moving Window MCEM.

There are two advantages over MCEM. Firstly, the cost of each update of the hyperparameters is constant and does not scale with $m$ since it only requires a single sample. Secondly, Moving Window MCEM converges faster than MCEM. The middle plot of Figure 3 demonstrates this. MCEM iteratively fits the hyperparameters for a specific set of posterior samples. Since hyperparameters and posterior samples are highly coupled, this alternating update scheme converges slowly [Neath et al., 2013]. To mitigate this problem, Moving Window MCEM continuously updates its population of samples by generating a new sample after each gradient step.

To produce the plot in the center of Figure 3, we plotted the predictive log-likelihood on the test set against the number of iterations of the algorithm to demonstrate the superior performance of Moving Window MCEM over MCEM. For MCEM, we used a set size of $m = 10$ (larger $m$ would slow down the method) which we generated over 500 MCMC steps. For Moving Window MCEM, we used a window size of $m = 300$. The model used in this experiment is a DGP with one hidden layer trained on the *kin8nm* dataset.

## 5   Decoupled Deep Gaussian Processes

This section describes an extension to DGPs that enables using a large number of inducing points without significantly impacting performance. This method is only applicable in the case of DSVI, so we considered it as a baseline model in our experiments.

Using the dual formulation of a GP as a Gaussian measure, it has been shown that it does not necessarily have to be the case that $\tilde{\mu}$ and $\tilde{\Sigma}$ (Eq. 1) are parameterized by the same set of inducing points [Cheng and Boots, 2017, 2016]. In the case of DGPs, this means that one can use two separate sets of inducing points. One set to compute the layerwise mean and one set to compute the layerwise variance.

In the variational inference setting, computing the layerwise variance has a significantly higher cost than computing the layerwise mean. This means that a larger set of inducing points can be used to compute the layerwise mean and a smaller set of inducing points to compute the layerwise variance to improve the predictive performance without impacting the computational cost.

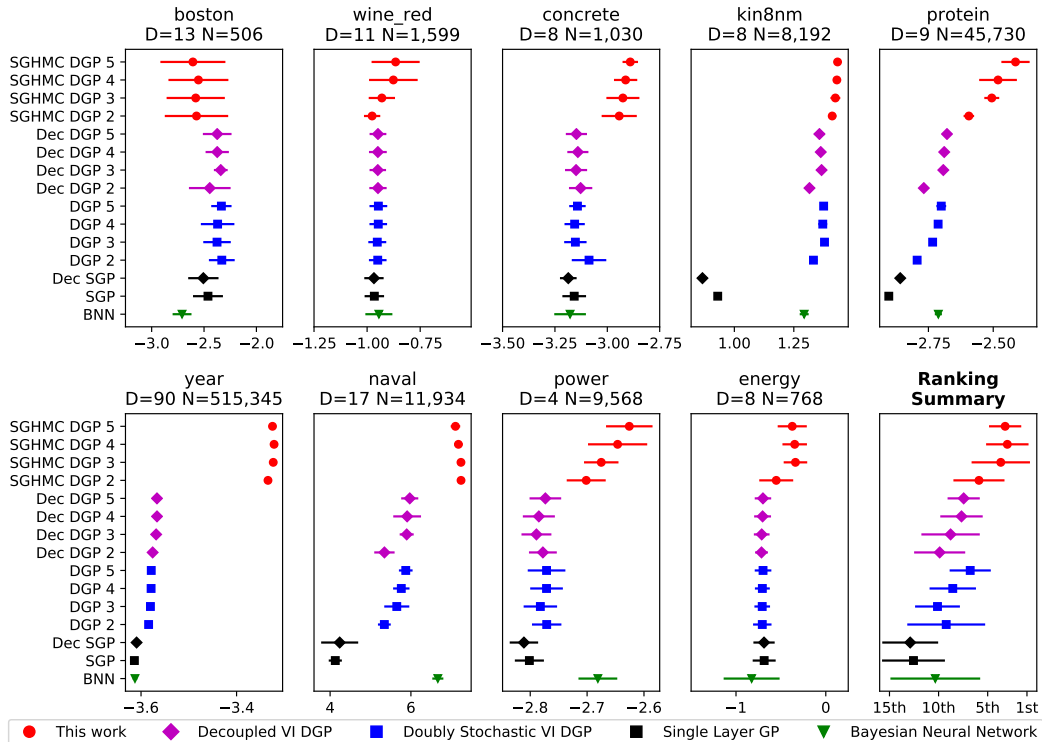

Figure 4: Log-likelihood and standard deviation for each method on the UCI datasets. Ranking Summary: average rank and standard deviation. Right is better. Best viewed in colour.

Unfortunately, the parameterization advocated by Cheng and Boots [2017] has poor convergence properties. The dependencies in the ELBO result in a highly non-convex optimization problem, which then leads to high variance gradients. To combat this problem, we used a different parameterization that lifts the dependencies and achieves stable convergence. Further details on these issues can be found in the supplementary material.

## 6   Experiments

We conducted experiments[2] on 9 UCI benchmark datasets ranging from small ($\sim$500 datapoints) to large ($\sim$500,000) for a fair comparison against the baseline. In each regression task, we measured the average test Log-Likelihood (MLL) and compared the results. Figure 4 shows the MLL values and their standard deviation over 10 repetitions.

Following Salimbeni and Deisenroth [2017], in all of the models, we set the learning rate to the default 0.01, the minibatch size to 10,000 and the number of iterations to 20,000. One iteration involves drawing a sample from the window and updating the hyperparameters by gradient descent as illustrated in Algorithm 1 in the left side of Figure 3. The depth varied from 0 hidden layers up to 4 with 10 nodes per layer. The covariance function was a standard squared exponential function with separate lengthscales per dimension. We exercised a random 0.8-0.2 train-test split. In the *year* dataset, we used a fixed train-test split to avoid the 'producer effect' by making sure no song from a given artist ended up in both the train and test sets.

**Baselines:** The main baselines for our experiments were the Doubly Stochastic DGPs. For a faithful comparison, we used the same parameters as in the original paper. In terms of the number of inducing points (the inducing inputs are always shared across the latent dimensions), we tested two variants. First, the original, coupled version with $M = 100$ inducing points per layer (DGP). Secondly, a decoupled version (Dec DGP) with $M_a = 300$ points for the mean and $M_b = 50$ for the variance.

These numbers were chosen so that the runtime of a single iteration is the same as the coupled version. Further baselines were provided by coupled (SGP: $M = 100$) and decoupled (Dec SGP: $M_a = 300$, $M_b = 50$) single layer GP. The final baseline was a Robust Bayesian Neural Network (BNN) [Springenberg et al., 2016] with three hidden layers and 50 nodes per layer.

**SGHMC DGP (This work):** The architecture of this model is the same as the baseline models. $M = 100$ inducing inputs were used to stay consistent with the baseline. The burn-in phase consisted of 20,000 iterations followed by the sampling phase during which 200 samples were drawn over the course of 10,000 iterations.

**MNIST classification**    SGHMC is also effective on classification problems. Using the Robust-Max [Hernández-Lobato et al., 2011] likelihood function, we applied the model to the MNIST dataset. The SGP and Dec SGP models achieved an accuracy of 96.8 % and 97.7 % respectively. Regarding the deep models, the best performing model was Dec DGP 3 with 98.1 % followed by SGHMC DGP 3 with 98.0 % and DGP 3 with 97.8 %. [Salimbeni and Deisenroth, 2017] report slightly higher values of 98.11 % for DGP 3. This difference can be attributed to different initialization of the parameters.

**Harvard Clean Energy Project**    This regression dataset was produced for the Harvard Clean Energy Project [Hachmann et al., 2011]. It measures the efficiency of organic photovoltaic molecules. It is a high-dimensional dataset (60,000 datapoints and 512 binary features) that is known to benefit from deep models. SGHMC DGP 5 established a new state-of-the-art predictive performance with test MLL of $-0.83$. DGP 2-5 reached up-to $-1.25$. Other available results on this dataset are $-0.99$ for DGPs with Expectation Propagation and BNNs with $-1.37$ [Bui et al., 2016].

**Runtime**    To support our claim that SGHMC has a lower computational cost than DSVI, we plot the test MLL at different stages during the training process on the *protein* dataset (the right plot in Figure 3). SGHMC converges faster and to a higher limit than DSVI. SGHMC reached the target 20,000 iterations $1.6$ times faster.

## 7    Conclusions

This paper described and demonstrated an inference method new to DGPs, SGHMC, that samples from the posterior distribution in the usual inducing point framework. We described a novel Moving Window MCEM algorithm that was demonstrably able to optimize hyperparameters in a fast and efficient manner. This significantly improved performance on medium-large datasets at a reduced computational cost and thus established a new state-of-the-art for inference in DGPs.

## Acknowledgements

We want to thank Adrià Gariga-Alonso, John Bronskill, Robert Peharz and Siddharth Swaroop for their helpful comments and thank Intel and EPSRC for their generous support.

Juan José Murillo-Fuentes acknowledges funding from the Spanish government (TEC2016- 78434-C3-R) and the European Union (MINECO/FEDER, UE).

## Footnotes

[1]Image source: Daniel Hernández-Lobato

[2]Our code is based on the Tensorflow [Abadi et al., 2015] computing library and it is publicly available at `https://github.com/cambridge-mlg/sghmc_dgp`.

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
