[Reviews · NeurIPS 2018]

Reviewer 1



Update after rebuttal: I think the rebuttal is fair. It is very reassuring that pseudocode will be provided to the readers. I therefore keep my decision unchanged. Original review: In the paper "Inference in Deep Gaussian Processes using Stochastic Gradient Hamiltonian Monte Carlo" the author(s) consider the problem of inference for deep gaussian processes (DGPs). Given the large number of layers and width of each layer, direct inference is computaitonal infeasible, which has motivated numerous variational inference methods to approximate the posterior distribution, for example doubly stochastic variational inference (DSVI) of [Salimbeni and Deisenroth, 2017] The authors argue that these unimodal approximations are typically poor given the multimodal and non-Gaussian nature of the posterior. To demonstrate these characteristics, the author(s) using an MCMC approach (specifically SGHMC of [Chen et al, 2016]) to sample from the posterior, thus providing evidence of the non-Gaussianity. Following this approach, they demonstrate the multimodality of a DGP posterior arising The disadvantage of the SGHMC algorithm is the large number of hyper-parameters to be tuned. The author(s) adopt a BNN approach and furthermore a Monte Carlo Expectation Maximisation approach (Wei, Tanner, 1990). The authors introduce a variant where instead of choosing a new set of samples each M step, the authors recycle the previous, replacing the oldest sample with a newly generated one (calling it moving window MCEM). To demonstrate their algorithm, they compare a number of non-linear regression algorithms on a set of standard UCI benchmark datasets, MNIST (using robust-max likelihood for classification), and Harvard Clean Energy Project dataset. I first qualify this review by saying that I am not an expert in deep gaussian processes, or methods for variational inference of DGPs, and so I cannot comment on the relative merits of these with any confidence. It is not surprising that VI are not able to capture the multimodality present in DGP posteriors, and it is good that this important feature has been identified. Certainly, if the DGP is a subcomponent of a larger Bayesian model, then failure to capture all modes of the posterior could be disastrous. Therefore, I definitely see the value of this proposed approach, and so I commend the authors for producing a well presented case for this. Two major concerns with this work relate to: (1) the lack of acknowledgement of similar works which apply MCMC for inference for DGPs , (2) the lack of clarity on certain aspects of the implementation of the SGHMC and the MW-MCEM method. I detail the issues in the following: 1. The paper [Hoffman, Matthew D. "Learning deep latent Gaussian models with Markov chain Monte Carlo." International Conference on Machine Learning. 2017.] also presents a stochastic gradient approach based on HMC. Although the context is slightly different, I feel it is similar enough that this should have been heavily referenced in this work, and formed the basis of the numerical comparison. Also related, to a lesser extent are [Han, Tian, et al. "Alternating Back-Propagation for Generator Network." AAAI. Vol. 3. 2017] and [Wolf, Christopher, Maximilian Karl, and Patrick van der Smagt. "Variational Inference with Hamiltonian Monte Carlo.", 2016]. 2 . Another major issue is the lack of details of the MCMC algorithm. Mainly it is absolutely not clear what form the subsampling takes. The author(s) refer to a "reparametrization trick to the layer outputs" citing Chen at al. 2014 without explaining further. Firstly, I'm not sure in what context [Chen et al 2014] use the parametrization trick. I usually refer to [D. J. Rezende, S. Mohamed, and D. Wierstra. Stochastic Backpropagation and Approximate Inference in Deep Generative Models. International Conference on Machine Learning, 2014.] and [D. P. Kingma, T. Salimans, and M. Welling. Variational Dropout and the Local Reparameterization Trick. 2015.] In particular, given that exact sampling is possible with cubic cost wrt N, it is important to understand the relative cost of iterating the SGHMC algorithm, i.e. the computational cost associated with producing a stochastic gradient as a function of N, minibatch size, etc. This has not been addressed in the paper. 3. The author(s) are correct that choosing m in the MCEM is a challenge. Too small risks overfitting of parameters, too large is expensive. I feel that the author(s) might not be aware of the well established literature on this problem. Indeed, this problem has been addressed in the statistics literature: One classical paper (Levine, Casella 2001) addresses the problem of recycling samples for the M step in some detail, proposing a further importance sampling step which would appear to be highly relevant to this situation.

Reviewer 2



This paper shows how inference in Deep Gaussian processes (DGPs) can be carried out using an MCMC approach. The approach proposed is not vanilla MCMC, but contains a novel way of considering samples which is more efficient and is claimed to prevent overfitting. While most of the recent DGP papers have focused on variational inference for approximating the intractable posterior, the present paper attacks the problem using MCMC. I find this approach to be particularly significant, because: (a) at least in small problems, we can get a feeling of how the exact posterior behaves (in contrast to variational inference that does not tell us how far the bound is from the real evidence). This, in turn, allow us to study DGP properties better. (b) the paper also demonstrates very good predictive capabilities of the new construction, which is an additional (if not the main) benefit. (c) I know many researchers in the field have thought about applying MCMC in DGPs to see what this alternative inference would achieve, but either almost no-one actually tried it "properly" or they got stopped by the highly correlated samples in the high-dimensional space; so it's great to finally have this new inference tool for DGPs. However, I think the recent approach of Dunlop et al. 2017 should be discussed and referenced. It is not the same approach taken here, but it is still an MCMC approach for sampling u given y in DGPs. With regards to novelty and technical merit, this paper: (a) starts from a good motivation (alternative inference for DGPs without the disadvantages of VI) (b) makes good observations (MCMC doesn't have to scale badly if we're clever about how to do it, e.g. lines 61-64 and the moving-window algorithm) (c) is correct and experimentally seems to work well. Concerning clarity and readability, I am a little conflicted but mostly positive overall. On the one hand, the paper reads nicely and explanations are extremely clear and concise and claims are balanced. On the other hand, I feel that some of the space in the paper is taken up by text that could be omitted (or put in Appendix) in favour of more important pieces. Specifically, Section 3 describes (in a clear a nice way nevertheless!) the multi-modal posterior aspect of DGPs which is basically common knowledge in this sub-field. Further, section 5 seems quite irrelevant to the main point of the paper, especially since the decoupling method cannot be applied to the MCMC case. In place of these parts, I'd prefer to see stronger theoretical analysis for moving-window MCEM (see below) or more plots from the experiments (see below). Overall, the presentation of this paper can be improved but it is nevertheless quite good. As mentioned above, it would be great to see further anlysis for MW-MCEM to make it more convincing and understandable. I understand that, compared to MCEM, the new approach changes samples continuously, however, the changes are in a more constrained way, so it is not clear to me why this would prevent overfitting or improve convergence speed. The analysis of MW-MCEM could also be isolated as it is the main methodological novelty of the paper, that is, I would prefer to see MW-MCEM tested additionally in a more general sampling setting instead of section 5. For example, MW-MCEM could be tested against Hensman et al. 2015 even for the shallow GP. Finally, I find the experiments conducted well and adequately. It would be interesting to see an explanation about the typically larger std for the MCMC method in Fig. 4 (I would actually expect the opposite to happen due to better exploration of multiple modes and therefore more consistent performance across runs, but maybe it's the number of samples that needs to be increased). It would also be great to have some accuracy/F1 measures to complement the LL results in Figure 4 (since experiments are run anyway, the authors can have this metric for free, so why not report it). References: Dunlop et al. 2017: How deep are deep Gaussian processes? ArXiv: 1711.11280 Other: - Are the DGP hyperparametres tuned in the 2nd phase of the algorithm (see lines 157-158) or not? - In references "gaussian" should be "Gaussian" - I think nowhere in the paper is mentioned explicitly what is "DGP 3/4/5", please explain with words for completeness. - Suggest to present MNIST results in a table. - Which data was used for the runtime plot? --- Update after rebuttal: I have read the authors' responses which address some of my questions. I think this is a good paper, but would further benefit from including discussion regarding the typically larger std that the MCMC method obtains in the experiments (this review point has not been addressed in the rebuttal).

Reviewer 3



The paper addresses the problem of doing inference in deep Gaussian processes (DGPs). The authors propose to use Stochastic Gradient Hamiltonian Monte Carlo (SGHMC) as a way to relax the assumption of Gaussianity for the inducing point posterior as assumed in the standard variational inference setting. To motivate this, the authors argue based on empirical evidence that the posterior distributions of interest are not always Gaussian nor unimodal. In order optimize the hyperparameters during training, they propose a new variant of the Markov Chain Expectation Maximization (MCEM) algorithm called Moving Window MCEM, which is claimed to perform better results than the original MCEM algorithm. Finally, the authors demonstrate the performance ofthe proposed method on a set of benchmark datasets (UCI datasets, MNIST classification, and Harvard Clean Energy). The experiments show that the proposed method outperforms the benchmark methods. The technical details of the submission appear sound and correct. While none of the claims in the paper is supported by any theoretical analysis, all claims are to some degree supported by empirical evidence. However, the claim that Moving Window MCEM performs better than MCEM is only supported by a plot of the training log likelihood vs run time for a single data set. Similarly, the claim that SGHMC converges faster than the reference method (DSVI) is also only supported by a plot of the test log likelihood vs run time for a single data set. However, the experiments based on the UCI datasets do show that the proposed method achieved better predictive performance than the benchmark methods, especially for the larger datasets. The paper is in general clear, well-organized, and well-written. However, the proposed method has many moving parts (SGHMC, the autotuning method for the SGHMC parameters and the Moving Window MCEM) and it’s not entirely clear how each of these is executed relative to each other. Therefore, it would be an improvement to include pseudo-code for the entire method in the supplementary material and not just for the Moving Window EM algorithm. Furthermore, there are a few statements that could be made more precise, see comments below. The combination of SGHMC and deep Gaussian processes is novel. Furthermore, the proposed variant of the MCEM is also original. Constructing deep models that are capable of quantifying uncertainty in a reasonable way is an important topic. Therefore, it is most likely that other researchers will use the proposed ideas and improve them. Additionally, the proposed method achieves state of the art performance in terms of predictive accuracy for a range of benchmark datasets and therefore practitioners will most likely also be interested in the method. Further comments: Line 61: “One might expect a sampling method to be more computationally intensive than an approximate method such as DSVI.” Sampling-based methods are also approximate Line 62: “However, in DGPs, sampling from the posterior is inexpensive, since it does not require the recomputation of the inverse covariance matrix, which only depends on the Hyperparameters.” The authors should clarify if the is always true or if it’s only true for the “sampling phase”, where the hyperparameters are kept fixed Line 82: “The probability of y...” This is not a meaningful quantity - the authors probably mean the conditional distribution of y Line 88: “The GP distribution f given the inducing” typo Line 126: “First, we illustrate with a toy problem that the posterior distribution in DGPs can be multimodal.” The authors state that there are two modes for the toy problem, but they should clarify of these modes are related to each other in any way. Furthermore, the plots in figure 1 only show f1 vs x and y vs f1, but it would be more interesting to see what y vs x looks like for the two modes. Line 140: "We examine the posterior samples generated by SGHMC for each inducing output" What measures have you taken to make sure that SGHMC have actually converged to the target distributions? If you cannot guarantee that the samples from SGHMC are representative of the true target distributions, then these tests lose their value. Line 194: Missing a log in the M-step? Line 212: “We plotted the predictive log-likelihood on the training set against the runtime of the algorithm to demonstrate the superior performance of Moving Window MCEM over MCEM (the test log-likelihood shows the same trend)” In that case, it would be more convincing to show the test log likelihood After the rebuttal ------------------------------ I have read the rebuttal, but it does not change my opinion of the paper as the authors did properly address all of my concerns. For example, it would have been nice if the authors would have commented on the nature of the multimodality in terms of symmetries and what measures they have taken to guarantee that SGHMC produces samples from the true posterior distribution for their statistical significance tests.